# Improved Workflow for Analysis of Vascular Myocyte Time-Series and Line-Scan Ca^2+^ Imaging Datasets

**DOI:** 10.3390/ijms24119729

**Published:** 2023-06-04

**Authors:** Madison Boskind, Nikitha Nelapudi, Grace Williamson, Bobby Mendez, Rucha Juarez, Lubo Zhang, Arlin B. Blood, Christopher G. Wilson, Jose Luis Puglisi, Sean M. Wilson

**Affiliations:** 1Lawrence D Longo MD Center for Perinatal Biology, School of Medicine, Loma Linda University, Loma Linda, CA 92373, USA; mboskind@students.llu.edu (M.B.); nnelapudi@students.llu.edu (N.N.); gewilliamson1@gmail.com (G.W.); bobbymendez@students.llu.edu (B.M.); lzhang@llu.edu (L.Z.); cgwilson@llu.edu (C.G.W.); 2Advanced Imaging and Microscopy Core, Department of Basic Sciences, School of Medicine, Loma Linda University, Loma Linda, CA 92350, USA; rjuarez@llu.edu; 3Department of Biostatistics, School of Medicine, California Northstate University, Elk Grove, CA 95757, USA; jose.puglisi@cnsu.edu

**Keywords:** smooth muscle, pulmonary vasculature, calcium, Ca^2+^ oscillation, Ca^2+^ sparks

## Abstract

Intracellular Ca^2+^ signals are key for the regulation of cellular processes ranging from myocyte contraction, hormonal secretion, neural transmission, cellular metabolism, transcriptional regulation, and cell proliferation. Measurement of cellular Ca^2+^ is routinely performed using fluorescence microscopy with biological indicators. Analysis of deterministic signals is reasonably straightforward as relevant data can be discriminated based on the timing of cellular responses. However, analysis of stochastic, slower oscillatory events, as well as rapid subcellular Ca^2+^ responses, takes considerable time and effort which often includes visual analysis by trained investigators, especially when studying signals arising from cells embedded in complex tissues. The purpose of the current study was to determine if full-frame time-series and line-scan image analysis workflow of Fluo-4 generated Ca^2+^ fluorescence data from vascular myocytes could be automated without introducing errors. This evaluation was addressed by re-analyzing a published “gold standard” full-frame time-series dataset through visual analysis of Ca^2+^ signals from recordings made in pulmonary arterial myocytes of en face arterial preparations. We applied a combination of data driven and statistical approaches with comparisons to our published data to assess the fidelity of the various approaches. Regions of interest with Ca^2+^ oscillations were detected automatically post hoc using the LCPro plug-in for ImageJ. Oscillatory signals were separated based on event durations between 4 and 40 s. These data were filtered based on cutoffs obtained from multiple methods and compared to the published manually curated “gold standard” dataset. Subcellular focal and rapid Ca^2+^ “spark” events from line-scan recordings were examined using SparkLab 5.8, which is a custom automated detection and analysis program. After filtering, the number of true positives, false positives, and false negatives were calculated through comparisons to visually derived “gold standard” datasets. Positive predictive value, sensitivity, and false discovery rates were calculated. There were very few significant differences between the automated and manually curated results with respect to quality of the oscillatory and Ca^2+^ spark events, and there were no systematic biases in the data curation or filtering techniques. The lack of statistical difference in event quality between manual data curation and statistically derived critical cutoff techniques leads us to believe that automated analysis techniques can be reliably used to analyze spatial and temporal aspects to Ca^2+^ imaging data, which will improve experiment workflow.

## 1. Introduction

Intracellular Ca^2+^ signals are well regarded for their regulation of cellular processes ranging from myocyte contraction, hormonal secretion, neural transmission, cellular metabolism, transcriptional regulation, and cell proliferation. Measurement of cellular Ca^2+^ is routinely performed using fluorescent microscopy techniques with biological indicators. Classic approaches rely on ratiometric fluorescent indicators including Fura-2 and Indo-1 that were originally designed roughly 40 years ago [1,2]. There are also single-excitation and emission indicators, such as Oregon Green, Fluo-4, and many other derivatives routinely used in confocal imaging studies of cells, complex tissues, and organ systems [3]. More recently, there has been the development of genetically encoded Ca^2+^ sensors, including those of the green fluorescent protein based GCaMP family, that are used in vivo, ex vivo, and in cellular preparations [4]. Regardless of the technique used, recording cellular Ca^2+^, by its nature, is performed in real time. For most intents and purposes, live cell Ca^2+^ studies are What-You-See-Is-What-You-Get. Experiments are often visceral and exciting with real-time gratification. Researchers can immediately recognize if they are recording real-time responses or operating with subpar preparations. However, moving from immediate qualitative responses to quantifiable data often presents challenges—even when using modern analytic approaches.

Over the last two decades, our research team has collected and published data using various fluorescent Ca^2+^ indicators in isolated cells and complex tissues. We have examined highly deterministic G-protein receptor-coupled Ca^2+^ signals and stochastic Ca^2+^ events that occur in random locations and time intervals during the data recording. Examination of deterministic Ca^2+^ signaling events due to agonist stimulation is relatively straightforward, as relevant data can be rapidly and easily identified based on the timing of cellular responses [5,6]. However, when performing quantitative analysis of stochastic events in complex tissues, our group estimates that we spend an order of magnitude more time performing image analysis than acquiring images at the microscope [7].

We initially analyzed Ca^2+^ signals by hand, which was very labor intensive. The workflow was similar to our electrophysiological studies [8,9], with manuscripts being based on relatively few experimental trials [10,11]. Over the past number of years our imaging and analysis workflow has become far more robust. Manuscripts are now based on hundreds if not thousands of data traces and events [7,12,13,14]. Analysis of these larger datasets is now performed using customized semi-automated analysis routines. Still, our research team has spent considerable time performing visual analyses of line scans (X,T) as well as videos (X,Y,T), ensuring the quality of each event. 

Performing such robust analysis and applying strict quality standards is critical to rigorous science; however, this is balanced by the amount of time required to analyze these large datasets. On smaller projects performing strict quality control checks is easily surmountable. Nevertheless, the time required to complete larger projects with thousands of line scans and events often requires hundreds of hours of visual analysis and associated quality checks by individual investigators. The time required for an observer to sift through large datasets can lead to experimenter fatigue, resulting in errors. Secondarily, even with oversight or blinded analysis by multiple observers and extensive training, there is the potential for experimenter error and bias. 

The purpose of the current study was to determine if we could expedite the image analysis workflow and overcome barriers that impede study completion and introduce experimenter error. This evaluation was addressed by re-analyzing a published “gold standard” dataset from recordings made in pulmonary arterial myocytes in en face arterial preparations [7], which were manually curated through visual analysis of the Ca^2+^ signals. For the current examination, we applied a combination of visually guided, data-driven, and statistical approaches to the datasets with comparisons to our published data to assess the fidelity of the various approaches.

## 2. Results

### 2.1. Ca^2+^ Oscillations

Zeiss LSM formatted files generated from confocal microscopy were previously analyzed via LCPro to detect cytosolic Ca^2+^ changes over time [7]. The parameter list generated by LCPro was then used to confirm Ca^2+^ activity through customized image J plug-ins and R script plotting programs. These routines provide digital printouts showing the coordinates for every detected region of interest (ROI) and time series tracings for the fluorescence intensity for each ROI. Traditionally, trained researchers visually check each time series trace to confirm whether ROIs represent an area of Ca^2+^ activity (true positive) and to remove any false positives (Figure 1).

For the purposes of the current study, results from the manually curated dataset generated in pulmonary vascular myocytes, which are broadly representative of smooth muscle myocytes of various origin [7] were compared to those using the various filters as defined in the Methods and performed on the raw LCPro dataset. Figure 2 shows the impact of data curation on the positive predictive value (PPV), sensitivity, and false discovery rate (FDR) for the various data filtering methods. These three measures were calculated based on the results of the true positives, false positives, and false negatives. The raw data and outlier fencing 5 method produces similar results. The data mined and outlier fencing 5 methods appeared to be the most accurate at filtering the data compared to the gold standard. The data mined method had a PPV and sensitivity of 74.6% and 98.1%, respectively. The outlier fencing 5 method had a PPV and sensitivity of 74.7% and 100%, respectively. Other data filtering methods performed with a PPV in the 75–80% range, but with marked decreases in the sensitivity as the filtering criteria became more strict. It is noteworthy that the sensitivity and PPV of the raw data (100% and 74.7%, respectively) are essentially equal to that of the data mined (98.1% and 74.6%) and outlier fencing 5 (100% and 74.7%) methods performed. The FDR was fairly similar between each of the filtering techniques, ranging from ~20 to 25%, except for the gold standard which had an FDR of 0%. Few statistical differences were found based on these data filtering approaches. Overall, this leads to the question of whether data filtration is critical since there seems to be no significant difference between the raw and gold standard values.

We then evaluated the impact of data curation and filtering on the quality of the Ca^2+^ oscillation events. Figure 3 shows summary results for the amplitude, area under the curve, rise time, decay, and duration, which were statistically examined to determine if there were any significant differences from one another or systematic data biases based on the filtration method used. Importantly, there were very few significant differences between the measured parameters and no systematic biases based on the data curation or filtering techniques.

### 2.2. Ca^2+^ Sparks

Zeiss LSM line scan files were analyzed with Sparklab 5.8 after conversion to TIFF. Sparklab 5.8 can either allow for the analysis of single files or automatically examine large batches of files. Notably, the detection processes associated with Sparklab 5.8 are slightly more robust than Sparklab 4.3.1, which was used for the analysis of the originally published dataset [7]. Unfortunately, the differences between Sparklab 4.3.1 and 5.8 preclude direct comparison of Ca^2+^ spark morphometry of spatial–temporal aspects of the events. Figure 4 shows a representative line scan with overlaid analysis showing manually observed Ca^2+^ spark events, denoted by the numbers, and those automatically detected by Sparklab 5.8 at various thresholds as indicated by the colored dots. The figure illustrates that low-amplitude events become increasingly more challenging to detect as the threshold is increased.

Figure 5 illustrates that the number of cells that were determined to be firing and the frequency of Ca^2+^ spark activity were sensitive to the chosen threshold. There was an incremental reduction in the number of identified events with a threshold of 0.5 identifying 595 spark events, 0.6 identifying 431 events, 0.7 identifying 382 events, 1.0 identifying 259 events, and 1.5 identifying 168 events. Raising the threshold also impacted the PPV, sensitivity, and false discovery rate, which was examined by direct observation in several traces from an adult normoxic animal. Generally, the PPV was greater than 90% at any of the thresholds examined in this dataset. The false positive rate was quite sensitive to the threshold examined, being nearly 10% at a threshold of 0.5. The false discovery rate dropped to ~5% at a threshold of 0.6 and above. The sensitivity of Sparklab 5.8 to find Ca^2+^ spark events was below 75%, even at the least stringent threshold of 0.5. The sensitivity incrementally decreased as the threshold was raised through the removal of lower-quality events that were not substantially out of the noise floor.

We then evaluated the impact of changing the threshold on quantitative response characteristics for Ca^2+^ sparks. The responses are shown in the scatter plots of Figure 6. The figure illustrates that changes in the threshold have minimal impact on Ca^2+^ spark amplitude, the whole width at half maximum, the time constant for decay (Tau), and the full duration at half maximum. Examination of the plots illustrates several events with spurious values, such as full width half maximum values that were over 100 microns or Tau’s, which were several seconds long.

## 3. Discussion

In this report, we describe the application of automated Ca^2+^ signaling analysis routines. We provide an introspective examination of procedures that are routinely used for the analysis of Ca^2+^ signals in cellular and tissue preparations using a previously published dataset that was manually curated by trained observers. The examinations we have performed on line-scan (2D; X,T) and video (3D; X,Y,T) recordings provide unique insight into the positive predictive value, sensitivity, and false discovery rates associated with these analysis procedures and the impact this has on the interpretation of results obtained through automated screening as compared to manually curated approaches. Table 1 provides an overview of the strengths and limitations associated with the use of LCPro and Sparklab 5.8 for Ca^2+^ signaling analysis along with a few salient recommendations, which can be broadly applied to analysis of Ca^2+^ signals made with other programs.

Data-driven and statistical-based screening techniques of spatial and temporal Ca^2+^ signals based on LCPro-detected events were informative. LCPro performs well on rounded cells such as the rat pulmonary microvascular endothelial cell monolayers that were examined in Francis et al., 2012 [15], and in our more recent studies performed in recordings of pulmonary arterial endothelial cells in en face preparations [6,16]. Over the last decade, we have published extensively using manually curated Ca^2+^ signaling datasets of vascular myocytes from LCPro. When we compare automated analysis processes to a “gold standard” dataset that was manually curated, we find that the interpretation of the findings is comparable with a positive predictive value of ~75%, a false discovery rate in the 25% range, and a sensitivity upwards of 95%. The PPV is low in large part because we are examining stochastic events that occur at random intervals during the recording. As compared to analyzing stochastic events, analysis of agonist-driven, deterministic events is rapid and easy as they can be filtered for true positives with negligible error by examining the timing of the event peak [5].

The current evaluation of an automated line-scan analysis procedure provides clarity for future studies. Our comparisons of various thresholds for automated processing provide insight into optimization strategies associated with Ca^2+^-spark detection and illustrate that our current detection strategies with Sparkmaster 5.8 are, at best, 70% effective, excluding low amplitude events that cannot be distinguished from background noise. Even still, use of Sparklab 5.8 yielded few false positives. The program has a positive predictive value of over 95%, and manually derived critical cutoffs based on our previous studies eliminate only a few additional outlier events. Thus, while Sparklab 5.8 does not detect every event, nearly all events that are detected are true positives. The inability of the Sparklab 5.8 to examine low amplitude sparks is noteworthy as these events may encode for activation of Ca^2+^-dependent signaling processes that remain poorly understood. This includes activation of Ca^2+^-sensitive ion channels, activation of cell signaling, and metabolic pathways [17,18,19]. 

Although the analysis procedures we are currently using are adequate for analyzing Ca^2+^ spark and oscillatory signals from vascular smooth muscle, a number of recently described automated analysis methods could provide greater precision. Open-source programs such as SIMA [20], DeepCAD [21], CITE-ON [22], CaImAn [23], and S8 [24] have been used for the detection of cells and Ca^2+^ signals in various imaging studies. There are also a range of deep-learning approaches being examined by various investigators for neuronal tracing in whole-brain studies, which provide additional toolsets and resources for cellular identification and image analysis [25]. In regard to Ca^2+^ signaling, these advancing approaches are well illustrated by DeepWonder, a recently designed program for analysis of wide-field Ca^2+^ imaging data from individual neurons in living brain [26]. Commercially available programs such as Visiopharm or Imaris can also be used to identify regions of interest and examine fluorescence intensity, and their toolkits are also expanding at a rapid pace. 

Automated detection and analysis of smooth muscle cells, which are long and narrow, as opposed to cuboidal or rounded cells, makes ROI detection challenging. The watershed algorithms used in currently available open-source and commercial programs have difficulties handling cylindrical as opposed to round objects. The issues with faithfully generating ROIs that properly outline smooth muscle cells that are elongated as opposed to round underlies sampling errors associated with LCPro [7,14]. The issues with ROI detection currently lead to oversampling, with multiple ROIs being generated in individual cells. Overcoming current limitations in ROI detection algorithms may require novel image analysis approaches. This may potentially include the use of wavelet analysis approaches [27,28], which break down image signals in both time and frequency domains. These wavelet analysis procedures greatly enhance edge detection, and thus, may be better suited to the detection of elongated smooth muscle cells, and other cylindrical or organically shaped cells, than commonly used watershed algorithms [29,30].

The current study was purposefully constrained to evaluating live-cell images made on vascular myocytes using confocal imaging of Fluo-4, a single excitation and single emission Ca^2+^ indicator [3]. The analysis processes of LCpro and Sparklab 5.8 were custom designed to detect events based on certain spatial and temporal characteristics on full-frame video and line-scan recordings, respectively. LCpro and Sparklab 5.8 each provide end users opportunities to modify the parameter set of the spatial–temporal detection system within certain constraints to optimize event detection. Notwithstanding performance issues in the highly complex arterial tissues of the current study, LCpro was designed for and is expected to function well with video files where the experimenter wishes to detect and analyze objects that have spatial and temporal increases in fluorescence intensity that meet specific criteria [15]. This would include videos of cellular fluorescence from various synthetic Ca^2+^ indicators such as Fluo-4, Oregon Green BAPTA-1, Cal-520 [31], and X-Rhod-1 along with genetically encoded Ca^2+^ indicators including those of the GCaMP family [32,33]. LCpro may also be of utility for analysis of ratiometric videos of commonly used dual excitation Fura-2 or dual emission Indo-1 indicators [34] or the newer generation indicator Asante Calcium Red [35], though to our knowledge LCpro has not been used for that purpose. Similarly, Sparklab 5.8 could be used for analysis of line-scan recordings made with a range of available synthetic Ca^2+^ indicators or genetic reporters [33]. 

Overall, the results of this introspective study regarding our current image analysis processes in vascular myocytes, where we compared automated and semi-automated analysis techniques to manually curated responses, are revealing. The automated Ca^2+^ imaging analysis processes provide a reasonable method that can be employed to rapidly screen for significant biological effects in small as well as large datasets. We would also submit that the data-driven analytical filtering techniques can be employed regardless of what specific analysis algorithms an experimenter chooses. This is especially true if the experimenter has access to curated datasets from previous studies that can be used as training sets to develop critical cutoffs. These procedures expedite analysis processes with reasonable fidelity, decreasing the time required for a typical analysis by an estimated 75–90%.

## 4. Materials and Methods

The data in this manuscript were previously published [7] and are based on experimental animal, tissue preparation, and confocal imaging studies as outlined in [5,7,36]. In brief, studies were conducted in accordance with the Animal Welfare Act, the National Institutes of Health Guide for the Care and Use of Laboratory Animals(National Research Council et al. 2011), “The Guiding Principles in the Care and Use of Animals” approved by the Council of the American Physiological Society. The work was conducted through animal use protocol 8,110,004 that was pre-approved by the Institutional Animal Care and Use Committee of Loma Linda University (IACUC-LLU) with an initial submission of 10 February 2011 and a final determination date of 15 March 2014.

### 4.1. Confocal Microscopy Studies

[Ca^2+^]_i_ was measured in pulmonary arterial myocytes in situ, using an en face preparation with the Ca^2+^ sensitive dye Fluo-4 AM (Invitrogen, Carlsbad, CA, USA) and a Zeiss 710 NLO laser scanning confocal imaging workstation (Carl Zeiss AG, Thornwood, NY, USA) with an inverted microscope (Zeiss Axio Observer) using procedures and processes as described [7]. In brief, a series of 500 images (~210 s each) were made to assess whole-cell Ca^2+^ oscillatory activity in individual cells, which was followed by 30–50 line scans of 18.9 s that were made to measure subcellular Ca^2+^ spark activity, with 1 line being made in each cell.

### 4.2. Ca^2+^ Oscillatory Signal Analysis

As described, regions of interest with Ca^2+^ oscillations were detected automatically post hoc using the LCPro plug-in for ImageJ, with analysis following the workflow of Figure 7 [7]. For presentation purposes, the fractional fluorescence intensity was automatically calculated using LCPro [7]. For the purposes of the current study, comparisons were made between manually curated data where a trained, single observer removed erroneous data to develop a “gold standard” dataset with supervision [7]. This gold standard was compared to the removal of erroneous data by statistical approaches or through the development of critical cutoff filters based on the evaluation of the gold standard. Before removal of erroneous data, the dataset was separated into three categories based on event duration: less than 4 s, between 4 and 40 s, and greater than 40 s [7]. Events under 4 s were found to be rapid subcellular Ca^2+^ responses due to the activation of ryanodine receptors, which are commonly known as Ca^2+^ sparks [7,37]. Data with a duration between 4 and 40 s were then filtered based on cutoffs obtained from multiple methods, including 80th, 90th, 95th, and 99th percentiles, outlier fencing, and data mining the gold standard data to create lower and upper critical cutoffs. The data mining technique was developed by examining the histogram and creating fences that contained logical Ca^2+^ events based on the rise time (between 0.78 and 30.0 s), decay (between 0.78 and 36.72 s), amplitude (between 1.25 and 3.7 F/F0), area (between 6.74 and 108.5 F*s/F0), and duration (between 4.7 and 40 s) of the response. All outlier fencing methods were based on the first and third quartiles (Q1, Q3) and the interquartile range (IQR). The lower inner fence was Q1—(n × IQR), while the upper outer fence was Q3 + (n × IQR), where n represents 1.5, 3.0, and 5.0. Percentiles, quartiles, and interquartile ranges were obtained via Prism 9.1 (Graphpad Software, Boston, MA, USA). After filtering the data set, each method was compared by determining the number of true positives, false positives, and false negatives and calculating positive predictive value (PPV), sensitivity and the false discovery rate (FDR). The false negative rate was calculated by determining the number of true positive events eliminated from the LCPro derived dataset by each filtering technique under the presumption that LCPro accurately identifies all Ca^2+^ oscillations. Further statistical analysis was also performed, as mentioned below.

### 4.3. Line Scan Analysis

Ca^2+^ sparks were recorded in pulmonary arterial myocytes as previously described [7]. Processing of the recorded line scans was conducted using the novel custom program Sparklab 5.8 built in the labview (National Instruments, Austin, TX, USA) GUI environment. Figure 8 depicts the workflow of Sparklab 5.8, which has advanced image normalization and background subtraction processes. The analysis identified Ca^2+^ spark events and characterized Ca^2+^ spark morphology and spatial–temporal features of the events. Analysis worked toward the goal of optimizing the threshold used for Ca^2+^ spark detection and generating critical cutoffs for automated analysis. We initially evaluated data mining techniques based on critical evaluation of over 20,000 Ca^2+^ spark events that were published in previous studies performed in pulmonary [7,37], uterine [12], and basilar arterial smooth muscle [14]. However, that data mining approach was not incorporated into this manuscript, as the boundary limits were based on earlier versions of Sparklab, which had different background subtraction approaches that affected spatial–temporal responses. Instead, we performed a robust evaluation of the threshold for Ca^2+^ spark responses above background noise using various thresholds for spark detection including 0.5, 0.6, 0.7, 1.0, and 1.5 above background noise. Notably, our previous projects using various versions of Sparklab have used a threshold of 0.5 for the detection of Ca^2+^ sparks [7,12,14,38], with visually guided extraction of false positive events. The resultant data derived from various thresholds were compared to visual guided analysis of selected line scans by a single observer with verification by a secondary observer. A total of 144 true positive Ca^2+^ spark events were manually observed in the line-scan traces, and comparisons were made to automated detection at various thresholds. From these data, the number of true positives, false positives, and false negatives were determined. This allowed for the calculation of the PPV, sensitivity, and FDR. The resultant analysis of Ca^2+^ spark events for event amplitude full width half maximum (FWHM), full duration at half maximum (FDHM), and the time constant for decay (Tau) were then filtered using our datamined criteria of amplitude (Lower and Upper), FWHM (Lower amd upper), FDHM (Lower and Upper), and Tau (Lower and Upper), and statistical comparisons made.

### 4.4. Chemicals Reagents and Drugs

The current studies were based on an analysis of existing data as described in [7]; thus, no additional chemicals or reagents were used in the re-analysis of the dataset.

### 4.5. Statistical Methods and Sampling

All time-series recordings were graphed and statistical analyses performed with Prism 9.1. Summarized data for the Ca^2+^ oscillatory signals are presented as violin plots with the median and 25 and 75% quartiles, while Ca^2+^ spark events are displayed as individual data points with mean and standard deviation. Data were evaluated for normality prior to any comparative statistical analysis. The specific test used is denoted in the figure legend or table including a Kruskal–Wallis One Way ANOVA with a Dunn’s multiple comparison test. *p* < 0.05 was considered statistically significant, unless otherwise noted. The number of studies performed are provided as the number of events, arteries, animals, or scans examined. The exact details are provided along with each figure. The percentage of cells firing with Ca^2+^ sparks was computed as the number of individual line scans with events relative to the total number of line scans, with statistical differences examined by Fisher’s exact test.

## Figures and Tables

**Figure 1 ijms-24-09729-f001:**
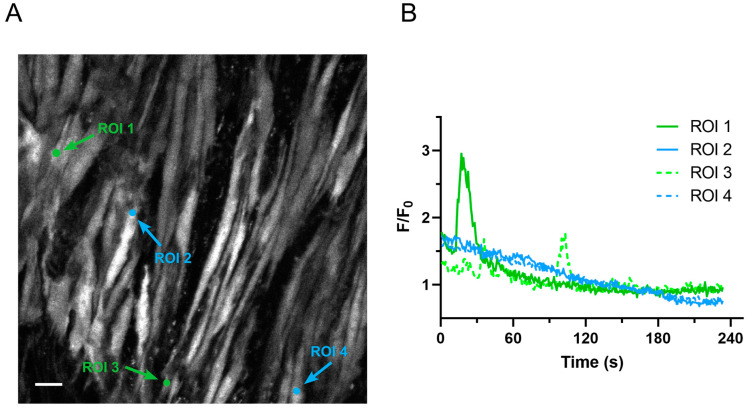
Representative true and false positive oscillatory Ca^2+^ events in recordings of pulmonary arterial myocytes from a hypoxic adult sheep recorded en face under control conditions. (**A**) Fluo-4 fluorescence of recorded cells at one time point using laser scanning confocal microscopy. Measurements of cytosolic Ca^2+^ changes over time were detected in ROIs using LCPro. Arrows point to regions of interest in individual myocytes. ROIs 1 and 3 represent areas of Ca^2+^ oscillations (true positives), while ROIs 2 and 4 are erroneous ROIs that were automatically detected (false positives). (**B**) Raw data traces for each of the four ROIs. Ca^2+^ oscillation recordings were made in arterial segments from 3 adult long-term hypoxic animals with a 63x water immersion c-Apochromat 1.4 NA objective at 1.28 Hz. F/F0: difference between peak fluorescence and background fluorescence. Scale bar 10 microns.

**Figure 2 ijms-24-09729-f002:**
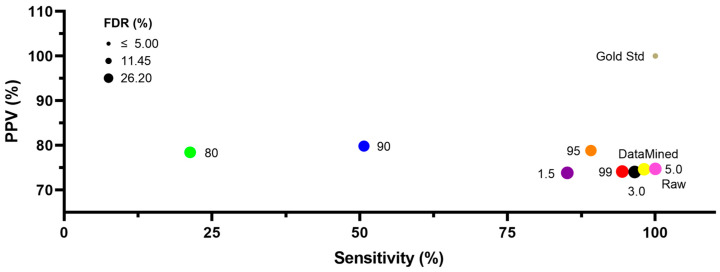
Performance comparison of filtering the Ca^2+^ oscillation training set obtained from hypoxic adult sheep pulmonary arteries under control conditions via eight different methods. In total, 99th (red), 95th (orange), 90th (blue), and 80th (green) percentiles as cutoffs, outlier fencing with an upper fence of 1.5 (purple), 3 (black), and 5 (pink) times the interquartile range, and through data mining (yellow) based on the gold standard. The sensitivities and positive predictive values (PPVs) for the gold standard (gold) and raw data (pink) are also present. The raw data and outlier fencing 5 method have the same values. Positive predictive value (PPV), sensitivity, and the false discovery rate (FDR) were calculated based on the results of the true positives, false positives, and false negatives from the LCPro datasets.

**Figure 3 ijms-24-09729-f003:**
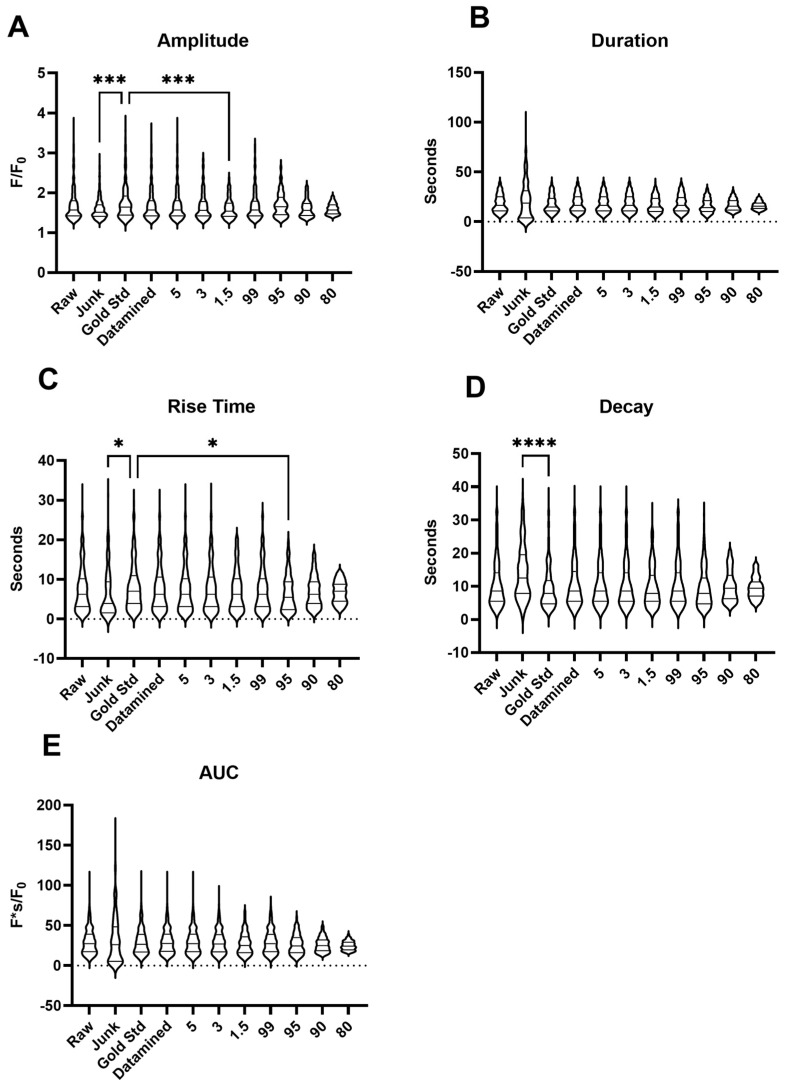
Data filtering to remove erroneous data has little influence on the kinetic measures of Ca^2+^ oscillations. Summary violin plots of various parameters across all comparisons for Ca^2+^ oscillations as determined through analysis by LCPro. (**A**) amplitude of the fractional fluorescence, (**B**) duration of the event, (**C**) rise time, (**D**) decay time in cytosolic fractional fluorescence, and (**E**) area under the curve. * *p* < 0.05, *** *p* < 0.001, **** *p* < 0.0001 indicates significance based on a Kruskal–Wallis ANOVA with a Dunn’s multiple comparisons test based on ranks as compared to the gold standard dataset as published in [7].

**Figure 4 ijms-24-09729-f004:**
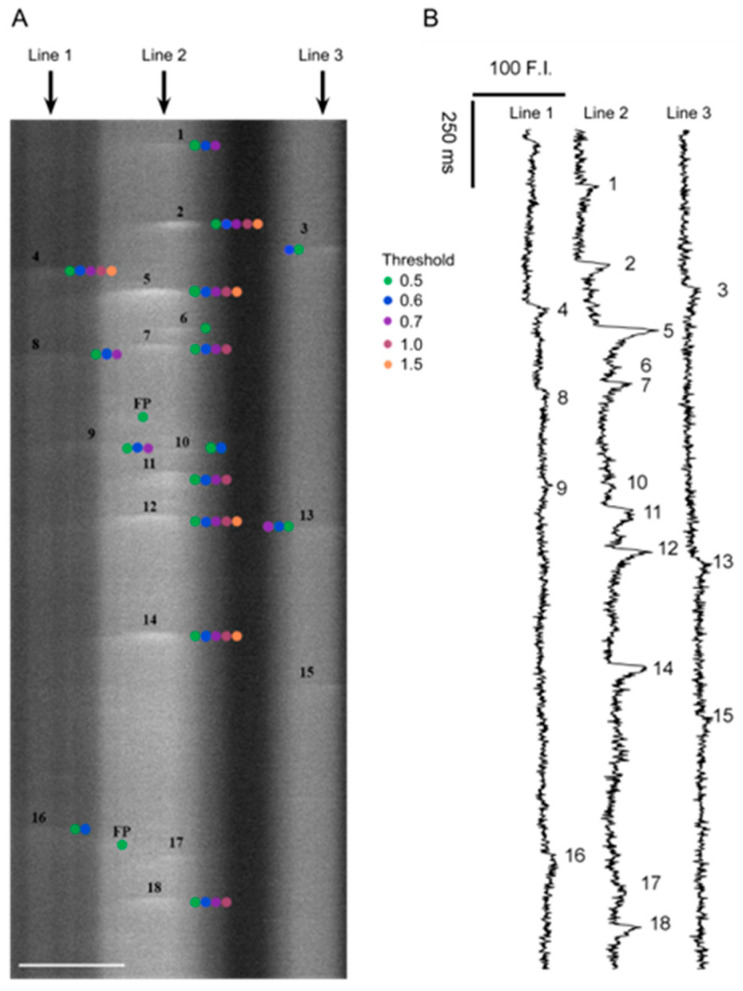
Representative line-scan recording in a pulmonary arterial myocyte from a normoxic sheep recorded en face under control conditions. (**A**) Fluo-4 fluorescence with manually and automatically detected Ca^2+^ spark events in a line scan recording. The numbers shown on the line scan correspond to manually detected events in the recording. The colored dots correspond to events detected at various thresholds by Sparklab 5.8 as provided in the legend. (**B**) Fluorescence intensity measurements over time corresponding to lines 1, 2, and 3 depicted in panel A. The numbers alongside each trace correspond to fluorescence intensity increases associated with Ca^2+^ spark events identified by a trained observer and presented in panel A. Scale Bar 5 microns.

**Figure 5 ijms-24-09729-f005:**
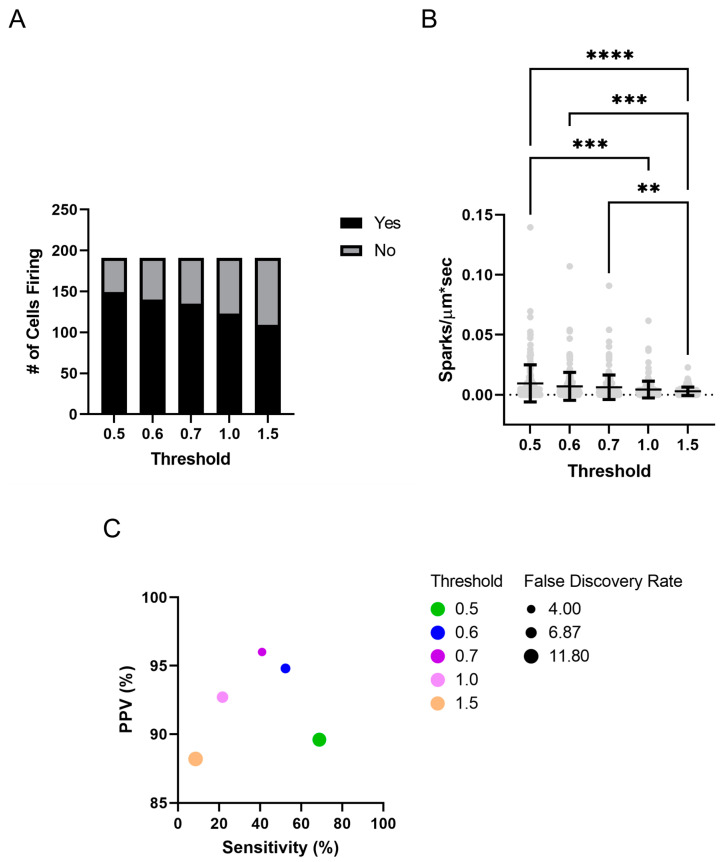
Performance comparison of changing threshold criteria on the Ca^2+^ spark training set obtained from normoxic adult sheep pulmonary arteries under control conditions. (**A**) Numbers of cells with or without detectable Ca^2+^ sparks at different thresholds. (**B**) Ca^2+^ spark firing frequency at various thresholds. (**C**) The sensitivity of identifying a Ca^2+^ spark as a function of the positive predictive value (PPV) along with the false discovery rate at various thresholds. Values are mean ± SD. Data were analyzed by a Kruskal–Wallis one-way ANOVA with Dunn’s multiple comparison test. ** *p* < 0.01, *** *p* < 0.001 **** *p* < 0.0001. Control responses were obtained from 192 lines from 4 animals.

**Figure 6 ijms-24-09729-f006:**
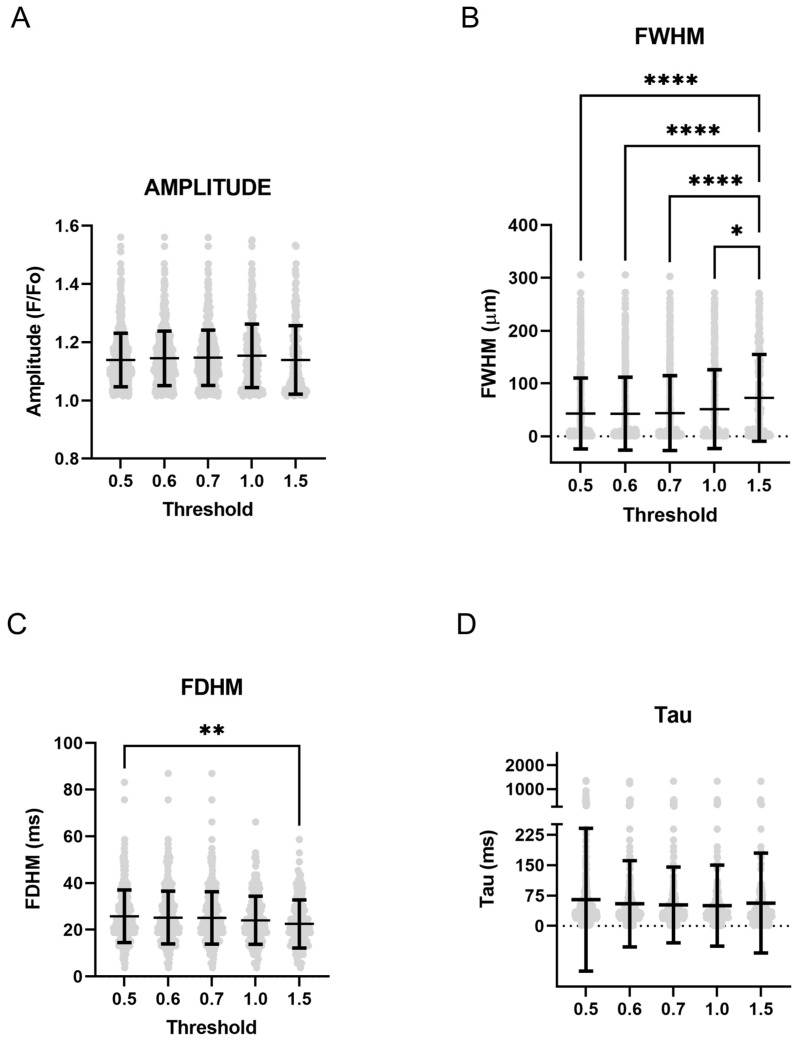
Changing threshold criteria has limited impact on spatial and temporal aspects of calcium sparks. (**A**) Ca^2+^ spark amplitude, (**B**) full width at half maximum (FWHM), (**C**) full duration at half maximum (FDHM), and (**D**) Tau. Values are mean ± SD. Data were analyzed by a Kruskal–Wallis one-way ANOVA with Dunn’s multiple comparison test. * *p* < 0.05, ** *p* < 0.01, **** *p* < 0.0001. Control responses were obtained from 192 lines from 4 animals.

**Figure 7 ijms-24-09729-f007:**
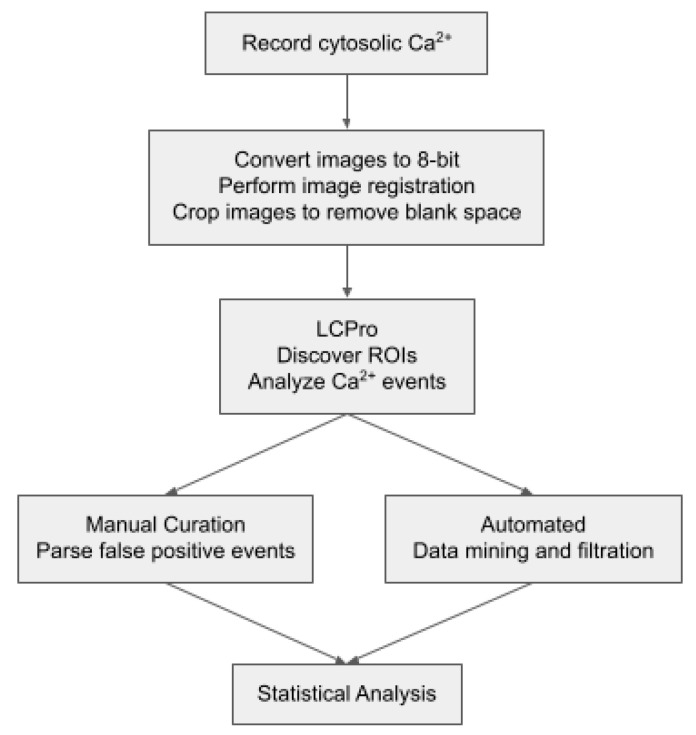
Whole-cell Ca^2+^ oscillatory signal analysis workflow. The schematic illustrates the process of collecting and analyzing the Ca^2+^ recordings along with the comparison between the manual curation process of Shen et al. 2018 [7] and the automated processes of the current studies.

**Figure 8 ijms-24-09729-f008:**
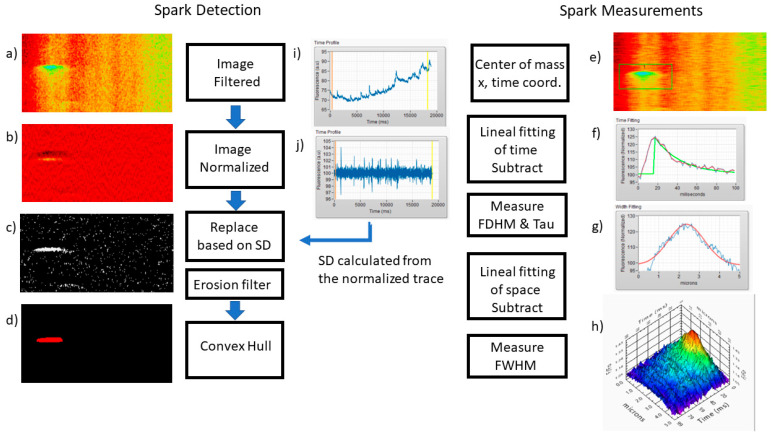
Line scan recordings of Ca^2+^ signals are analyzed using the following processes. (**a**) Filtration. A lineal filter is applied to the image with spatial and time cutoff set by the user, (**b**) Normalization. Each line is normalized in the time domain by dividing each point with the average value of five points before and after, (**c**) Replacement. The standard deviation (SD) is calculated from the normalized image and used as a criteria for replacement by a threshold “n” relative to the SD. Points above (n × SD) will be set to 1 otherwise 0, in the example analysis n was set to 1.5. An erosion filter is applied to the normalized image to suppress the “salt and pepper” noise. (**d**) Localization. A convex hull procedure is carried out to determine the location of potential sparks. Minimum area criteria is set to eliminate false positives events. (**e**) Centroid. The center of mass is used to localize the coordinates for maximum fluorescence. (**f**) On the time domain a lineal fit is used to equalize the beginning and end of the event. FDHM is calculated and an exponential fit (green line) is carried out to obtain the Tau. (**g**) Similar to the time domain, a lineal fit is used on the space dimension to equalize the left and right size of the spark. A gaussian fit is used to calculate the FWHM (red line). (**h**) Multidimensional rendering of the spark. (**i**) Raw fluorescence over time for the acquired image. Notice the changes in the baseline fluorescence. (**j**) After normalization, the fluorescence time profile will have a mean value of 1 with an SD. The SD is used as a criterion to accept and remove pixels from the image. Automation is performed by applying this process to all files in the folder using pre-set parameter sets and criteria, with pixel size and scanning rate metadata provided for each image file.

**Table 1 ijms-24-09729-t001:** Strengths and limitations of LC Pro and Sparklab 5.8.

**LC Pro**	
**Strengths**	**Limitations**
- Rapid analysis of Ca^2+^ imaging data from full-frame video recordings. - Image J plugin. - Performs well on rounded cells. - Performs well on stochastic datasets. - Can detect whole cell and subcellular events. - Ability to use varied sized “seeds” to aid detection. - Ability to set threshold criterion.	- Tends to have a high false positive rate. - Oversampling issues with oblong cells. - Anomalous findings require secondary data filtration or manual evaluation.
**Recommendation**	
- LC Pro should be evaluated with a training dataset to optimize event detection prior to analysis of large datasets. - Spatial and temporal data should be filtered either on statistical criteria, empirical upper and lower limits, or from the existing literature.	
**Sparklab 5.8**	
**Strengths**	**Limitations**
- Rapid detection of Ca^2+^ sparks from line-scan recordings. - Customized spatial and temporal settings. - Automated analysis of large datasets. - Low false positive rate. - LabVIEW runtime VI.	- Difficulty detecting small amplitude events. - Spatial and temporal analysis can have anomalous results.
**Recommendation**	
- Evaluate multiple thresholds and customize spatial–temporal settings in a training dataset to optimize sensitivity and specificity for event detection. - Spatial and temporal data should be filtered either on statistical criteria, empirical upper and lower limits, or from the existing literature.	

## Data Availability

Data are available upon request.

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
