# Peer review of "Improved Workflow for Analysis of Vascular Myocyte Time-Series and Line-Scan Ca2+ Imaging Datasets"

_ijms, 2023, doi:10.3390/ijms24119729_

Round 1
Reviewer 1 Report
Given the discrepancy of the two review reports on this manuscript and the relevance of the third comment to make a definitive decision, I admit that I cannot give a final conclusion concerning the quality of the paper. In my opinion the applied automated analysis techniques and the evaluation of its reliability obtained by re-analyzing pre- published “gold standard” datasets is not clearly described and is difficult to interpret.
Author Response
We hope that the reviewer finds the revisions that were made to address reviewer 3 and the academic editor acceptable.
Reviewer 2 Report
The manuscript "Improved workflow for analysis of 2D and 3D Ca2+ imaging datasets" reports on a comparison among manual and automated analysis techniques for Ca2+ imaging studies. Different parameters have been considered both on 2 and 3D datasets and statistical differences have been reported. The authors conclude that manual and automated analysis techniques do not differ in a significant manner, suggesting that automated analysis may improve experiment workflow.
No major criticism are present in the text, experimental design and data reporting.
Line 119- 120: please provide the extended term for PPV and FDR.
Author Response
Corrections have been made to include positive predictive value (PPV) and false discovery rate (FDR).
Reviewer 3 Report
The authors have systematically compared the outcomes of different approaches to analyze Ca2+ signals from a specific dataset. They were specifically interested in how well automated data curation techniques matches the outcomes of visually assessed recordings. In the view of the increasing amounts of data the researchers have to face in their analyses and the time that can be saved by automating these rather time-consuming tasks in of high relevance. The approaches that the authors use to compare the performance of different types of analysis seems good and the manuscript is well-written. However, despite these positive sites I do not recommend publication of this manuscript in International Journal of Molecular Sciences for the following reasons:
The authors focus the existing recordings of a specific cell type with specific signals, i.e., pulmonary arterial myocytes. The conclusions drawn from their study are therefore not general in hence limited applicability.
Several other important aspects referring to different technical aspects of Ca2+ recordings, such as the spatio-temporal resolution, the role of different dyes, etc., are not considered, even though this aspects are highly relevant in the context of such assessments.
The authors have not developed a novel and advanced methodological tool but use established tools and evaluate its effectiveness. While this is in principle okay, it does not provide ground-breaking methodological concepts that could drastically improve the time-consuming task of Ca2+ signal analysis across different research groups.
In sum, while the paper has some strong points and that are relevant for researchers dealing with Ca2+ signalling analysis in myocytes, due the limited applicability, the lack of generality, and the limited methodological advancement, I do not recommend its publication in a high-impact journal like IJMS.
Round 2
Reviewer 3 Report
The authors have somewhat supplemented their manuscript and provided some additional information. However, the major issue, i.e., the work does not represent a major advancement, remains. As I have highlighted in my first round of review – evaluating methodological toolkits is of interest and deserves to be published, but as this work does not represent a ground-breaking advancement that would be of interest to the wider audience, a more specialized technical journal would be the more suitable venue for this work.